# A simplified method for evaluating swallowing ability and estimating malnutrition risk: A pilot study in older adults

**Nareudee Limpuangthip[1]☯, Orapin Komin[1]☯\*, Teerawut Tatiyapongpaiboon[2]**

1 Department of Prosthodontics, Faculty of Dentistry, Chulalongkorn University, Bangkok, Thailand,
2 Thungyai Hospital, Nakhon Si Thammarat, Thailand

☯ These authors contributed equally to this work.
\* Orapin.geriatric@gmail.com

**Data Availability Statement:** All relevant data are within the manuscript and its Supporting information files.

## Abstract

### Objectives

The aim of this pilot study was to develop a Thai-version of a simple swallowing questionnaire, called the T-SSQ, and to evaluate the association between malnutrition risk and swallowing ability, determined objectively by tongue strength and subjectively by the T-SSQ. Sensitivity analysis was also performed to determine which swallowing indices better estimate malnutrition in older adults.

### Methods

This cross-sectional study comprised two phases: Phase I, development and cross-cultural translation of the T-SSQ; and Phase II, application of the T-SSQ in 60 older adults. In Phase I, content and face validity of the T-SSQ was evaluated by 10 experts and 15 older adults. In Phase II, the convergent validity of the T-SSQ was evaluated by determining its association with objective tongue strength. Nutritional status was evaluated using the Thai-version of the Mini-Nutritional Assessment. Covariates included sociodemographic characteristics, and oral and health-related status. Adjusting for covariates, the associations between the two swallowing indices and malnutrition risk were determined using multivariable regression analyses. A cut-off value for low tongue strength was determined using a receiver operating characteristic (ROC) curve, and sensitivity analysis between the swallowing indices and malnutrition risk was performed.

### Results

The T-SSQ comprised 4-items of common signs and symptoms of a swallowing problem. Its content and face validity were verified. Older adults were considered as having a swallowing problem when at least one item was reported. Convergent validity of the subjective index was shown by significantly different tongue strength values between the participants with and without a swallowing problem (p for independent t-test = 0.014). Based on the highest area under the ROC curve, an 18-kPa cut-off value was chosen to classify low tongue

**Funding:** Yes. This research is funded by Chulalongkorn University, Grant number CU_GR_63_11_32_04. The funders had no role in study design, data collection and analysis, decision to publish, or preparation of the manuscript.

**Competing interests:** The authors have declared that no competing interests exist.

strength. Having a swallowing problem and low tongue strength was significantly associated with malnutrition risk. The positive predictive value of the subjective swallowing index was 1.8-fold higher than objective tongue strength.

## Conclusions

Self-reported swallowing problems determined by the T-SSQ can be used as a subjective index for evaluating swallowing ability in older adults. Subjective swallowing problems and objective tongue strength were associated with malnutrition risk. However, the T-SSQ estimated malnutrition risk better than the objective index.

## Introduction

Oral and general health functionally decline as people age [1]. Gradually declined oral function can lead to oral frailty followed by oral hypofunction. However, they can recover to the healthy stage by early detection and proper dental treatment [2]. Oral health becomes oral frailty when a person has decreased occluding pairs of natural teeth, increased unchewable foods, or slight choking/spillage while eating. Moreover, oral hypofunction is diagnosed when 3 out of 7 oral signs or symptoms are present: oral uncleanness and dryness, reduced occlusal force, declined masticatory function, reduced tongue and lip motor function, and reduced tongue pressure and swallowing function [2]. Because eating and swallowing ability plays a major role in oral function, a decline in swallowing ability contributes to malnutrition [3, 4]. Malnutrition increases the risk of morbidity and mortality, and negatively affect the quality of life of older adults [5].

To prevent malnutrition in older adults, early detection of declined swallowing ability is necessary. Several objective and subjective indices have been used to evaluate swallowing ability in older adults. Tongue pressure measurement is commonly used to objectively evaluate swallowing ability, because tongue motor function plays an important role in mastication and swallowing [3, 6]. However, this method requires special instruments and time to perform. To evaluate swallowing ability subjectively, the 10-item Eating Assessment Tool (EAT-10) is commonly used because it is considered a reliable and validated questionnaire [2, 6, 7]. However, some studies reported the limitations of EAT-10 regarding its substantial floor effect, several redundant items, and relatively low construct validity [8, 9]. Thus, an alternative simple screening method for evaluating swallowing ability should be proposed for the early detection of oral function when a patient is in the frailty or hypofunction stage.

The aims of the present study were to develop a Thai-version of a simplified swallowing questionnaire (T-SSQ), and to evaluate the association between malnutrition risk and swallowing ability, determined objectively by tongue strength and subjectively by the T-SSQ. In addition, sensitivity analysis was performed to determine which swallowing indices better estimate malnutrition in older adults.

## Materials and methods

### Study design and participants

The present study was a cross-sectional design. The study protocol was approved by the Ethics Committee of the Faculty of Dentistry, Chulalongkorn University (HREC-DCU 2018–112).

The participants and their guardians provided written informed consent prior to participating in the study.

The participants were older adults (aged $\geq$ 60 years old) recruited from patients who received dental treatment at the Geriatric and Special Patients Care Clinic, Faculty of Dentistry, Chulalongkorn University during August 2017– January 2019 (total duration of 1 year and 5 months). The exclusion criteria were patients who declined or were unable to perform a tongue pressure test due to severely declined functional or intellectual conditions, or currently had malnutrition.

## Phase I. Development of simplified swallowing questionnaire

A 9-item swallowing questionnaire (SQ) was created based on the common signs and symptoms of dysphagia patients according to Walker et al. (1990) [10] and Nawaz S & Tulunay-Ugur OE (2018) [11] (S1 File). Cross-cultural translation of the SQ was performed according to the WHO guidelines [12]. A forward translation from English to the Thai version was conducted by two-independent translators, one dentist and one non-dentist, and integrated into a single Thai version. A back translation from Thai to English version was then conducted by two-independent translators, one was a dentist and the other was not. To verity the content validity of the translated version, the principal investigators and translators discussed any discrepancies related to the meaning of words and phrases between the Thai and English versions. The investigators ensured that the basic concepts and meanings of all terms with reference to the original version were maintained, and then, the Thai-version of the swallowing questionnaire (T-SQ) was proposed.

The content validity of the T-SQ was evaluated by 10 experts (5 physicians and 5 dentists). They gave responses whether they agreed that each of the 9 items indicated a swallowing problem (agree, disagree). The items with less than 80% agreement were excluded, and the final version was reduced to 4 items. A face validity of the 4-item swallowing questionnaire was evaluated in 20 older adults (age range 52–77 years). They gave responses whether they clearly understood the description of each item using a dichotomous scale (clearly understood and unclear), and provided suggestions if any rephrasing was necessary for easier understanding. All items received 90–100% clearly understood response but with some suggestions. After the discrepancies were discussed, a final version of the Thai-version simplified swallowing questionnaire (T-SSQ) was developed. The word liquid in the English version was translated into naam in the Thai version, which means water (noun) or related to liquid substance (adjective) for better understanding by Thais.

The T-SSQ comprised 4 items: 1) having problems swallowing certain food or liquids, or could not swallow at all, 2) coughing or choking when eating or drinking, 3) aspirations with liquids or solid food occurs, and 4) a sensation that food got stuck in the throat or chest. The response to each item was given as having (1) or not having (0) a problem. A person was defined as having a swallowing problem when at least one of items was present at least once a week during the past month (S2 File).

## Phase II. Application of the T-SSQ

Phase II, the T-SSQ application, was conducted as a pilot study in 60 older adults who did not participate in Phase I. Swallowing ability was assessed subjectively by the T-SSQ and objectively by tongue strength. Convergent validity of the T-SSQ was evaluated using objective tongue pressure as a reference. Sensitivity analysis was performed to determine which subjective and objective swallowing indices better estimate malnutrition in older adults.

## Nutritional assessment

Nutritional status was measured using the Thai-version of the Mini-Nutritional Assessment (MNA) with a score ranging from 0–30 (S2 File). The participants were categorized as having malnutrition risk when MNA score = 17–23.5, and being normal when the score ≥24. The MNA was used because it is a standardized and validated instrument in older adults [13, 14].

## Swallowing ability assessment

Subjective swallowing ability was evaluated using the T-SSQ developed in Phase I. The participants were interviewed with the assistance of their caregivers, if present. The participants were defined as having a swallowing problem when at least one of the T-SSQ items was reported. The inter-examiner reliability was determined using 15 patients at the patients' first evaluation visit. The test-retest reliability was evaluated by reinterviewing the 15 patients one week later. The weighted Kappa scores for the inter-examiner and test-retest reliability assessments were approximately 0.82 and 0.87, respectively.

The objective swallowing ability was evaluated through tongue pressure, measured using the JMS TPM-02 measurement device (JMS, Inc., Hiroshima, Japan), which consisted of a plastic catheter and a balloon. The participants sat in an upright position. The balloon was inserted into their oral cavity and placed on the anterior part of the palate with their lips and jaw closed, while the plastic catheter was held at the midpoint of the central incisors. The participants raised their tongue and pressed the balloon against the hard palate as hard as possible, and the maximum tongue pressure (kPa, kilopascal) was read. This procedure was done in triplicate with 5 min resting intervals, and the tongue strength (kPa) was calculated from the average value of the three measurements [15–17].

## Covariates

Information regarding biological factors, oral- and health-related status was recorded. Biological factors were an individuals' age and sex. A dentist evaluated oral status, comprising the number of remaining functional teeth (ranged from 0 to 28 teeth), number of posterior occluding pairs (ranged 0–8 pairs), and type of denture worn. If more than one type of denture was present, it was classified as the type with the higher number of tooth loss.

Health-related status covered the participants physical and psychological conditions: dependency status and cognitive status, respectively. The clinical frailty scale (CFS) was used to categorize dependency status into independent, semi-dependent, and dependent [18]. Cognitive function was evaluated using the Thai-version of the Mini-Mental State Evaluation (MMSE) (S2 File). With a score ranging from 0–30, the participants were considered as having mild cognitive impairment (MCI) when the score was below 18 and below 22 when they had attained at least primary and above primary education, respectively [19].

## Power analysis

A power analysis of the sample size was performed to determine whether the T-SSQ was a sensitive assessment tool for using as a key instrument in early diagnosis and detecting swallowing impairment among older adults. The results indicated that the tongue strength (mean ±s.d.) of the participants who reported a swallowing problem ($n_1 = 7$) and those who did not ($n_2 = 54$) were 16.8 ±10.6 kPa and 26.7 ±9.6 kPa, respectively. Using the two-independent means test, a power of 70% was calculated at 5% type I error.

## Data analysis

Descriptive statistics was performed to determine the percentage (%) and mean ±standard deviation (s.d.). Univariate analyses of the associations between related variables and having a swallowing problem were analyzed using the chi-squared test, whereas its association with tongue strength and MNA score were analyzed using either one-way ANOVA or independent t-test. Variables with p-value < 0.10 were included in the multivariable analyses. Adjusting for covariates, multivariable logistic and linear regression were used to determine the factors associated with the subjective and objective swallowing indices, and their associations with malnutrition risk. To determine the convergent validity, the tongue strength values of the participants with and without swallowing problem assessed by the T-SSQ were compared using the independent t-test. A receiver operating characteristic (ROC) curve was plotted to determine the area under the curve (AUC) in the malnutrition risk models; the higher the AUC, the better the model was able to distinguish between the participants with and without malnutrition risk. To categorize the low and high tongue strength, a cut-off value that gave the highest AUC value was chosen. For the sensitivity analysis, the positive predictive value (PPV), negative predictive value (NPV), sensitivity, and specificity between malnutrition risk and the two swallowing ability indices were calculated. The data were analyzed using STATA version 13.0 (StataCorp LP) at a 5% significance level.

## Results

The T-SSQ was developed as a subjective swallowing index, comprising 4-items of common signs and symptoms of a swallowing problem. The content and face validity of the T-SSQ was assessed. The characteristics of the participants attending in Phase II based on the subjective and objective swallowing indices, and malnutrition risk are shown in Table 1. Their mean ±s. d. age was 78.0 ±7.0 years old. Malnutrition risk was found in 18% of the participants, while the others were within normal limits. MCI was present in 90% of the semi- and dependent older participants. Oral status was associated with the subjective and objective swallowing indices, and malnutrition risk. The convergent validity of the T-SSQ was revealed as shown by a significant difference in tongue strength values between the participants with and without a swallowing problem (p for independent t-test = 0.014).

There were significant associations between the subjective and objective swallowing indices and malnutrition risk after adjusting for potential covariates (Table 2). Because there was collinearity between dependency status and MCI, the MCI variable was not included in the multivariable regression models. Based on the ROC curve, 18 kPa was chosen as a cut-off value to categorize the participants into low and high tongue strength because it gave the highest AUC value when plotting the curve between tongue strength and malnutrition risk (Fig 1).

Estimates of the PPV, NPV, sensitivity, and specificity are presented in Table 3. The sensitivity value indicated that 45.5% and 36.4% of older adults having malnutrition risk would have a swallowing problem and lower tongue strength, respectively. The PPV values indicated that the participants with a swallowing problem were 1.5–2 folds more likely to have malnutrition risk than those who had lower tongue strength.

## Discussion

The present pilot study developed the Thai-version of the simplified swallowing questionnaire (T-SSQ) as a subjective index to evaluate the swallowing ability in older adults. The convergent validity of the T-SSQ was verified using objective tongue pressure as a reference. The findings from this pilot study revealed an association between the subjective and objective swallowing indices and malnutrition risk. Sensitivity analysis demonstrated that the ability of the T-SSQ

**Table 1. Characteristics of the study participants.**

| Variables | Overall | Swallowing ability index | | Nutritional status |
|---|---|---|---|---|
| | distribution: | Self-reported swallowing problem by T-SSQ (Yes): | Maximum tongue pressure (kPa): | MNA score: |
| | % | % | mean (±s.d.) | mean (±s.d.) |
| Overall | 100.0 | 11.7 | 25.5 (±10.1) | 26.0 (±3.0) |
| Age (years): 60–69 | 15.0 | 11.1 | 34.5 (±8.7) | 25.0 (±3.8) |
| 70–79 | 40.0 | 4.2 | 26.8 (±9.3)* | 26.5 (±2.1) |
| 80 and above | 45.0 | 18.5† | 21.4 (±9.4)* | 26.0 (±3.3) |
| Sex: Male | 48.3 | 10.3 | 27.7 (±9.8) | 26.9 (±2.1) |
| Female | 51.7 | 12.9 | 23.5 (±10.1)† | 25.2 (±3.4) |
| **Health-related status** | | | | |
| Dependency status: Independent | 71.7 | 4.6 | 27.8 (±9.8) | 26.9 (±2.1) |
| Semi-dependent | 18.3 | 27.3* | 19.1 (±8.9)* | 25.1 (±2.5) |
| Dependent | 10.0 | 33.3* | 21.2 (±8.8)* | 21.1 (±3.9)* |
| MMSE score: Normal | 78.3 | 6.2 | 26.5 (±10.4) | 26.5 (±2.2) |
| Mild cognitive impairment | 21.7 | 30.8* | 22.1 (±8.6)† | 24.1 (±4.4) |
| **Oral status** | | | | |
| Natural teeth: 0 | 19.7 | 16.7 | 24.1 (±10.2) | 26.0 (±3.6) |
| 1–19 | 70.5 | 11.6 | 25.2 (±9.7) | 25.9 (±3.0) |
| 20 and above | 9.8 | 0.0 | 30.1 (±9.9) | 27.0 (±2.0) |
| Posterior occluding pairs: 0 | 76.7 | 13.0 | 24.5 (±10.2) | 26.1 (±3.0) |
| 1–3 | 13.3 | 12.5 | 32.4 (±10.7) | 25.9 (±1.8) |
| 4 and above | 10.0 | 0.0 | 26.7 (±9.6) | 25.4 (±4.5) |
| Denture type: No denture | 3.3 | 0.0 | 30.7 (±1.8) | 29.2 (±1.1) |
| Removable partial denture | 45.0 | 7.4 | 27.3 (±8.8) | 26.0 (±2.7) |
| Complete denture | 51.7 | 16.1 | 23.6 (±11.1) | 25.8 (±3.2) |
| **Swallowing ability** | | | | |
| Having swallowing problem: No | 11.7 | - | 26.7 (±9.6) | 26.0 (±2.8) |
| Yes | 88.3 | | 16.8 (±10.6)* | 23.0 (±2.7)* |

*p <0.05,

†p<0.10 determined by independent t-test or one-way ANOVA and post-hoc comparison test.

**Table 2. The associations between swallowing indices and related variables.**

| | Swallowing ability index | | Malnutrition risk: adjusted OR (95% CI) | |
|---|---|---|---|---|
| Variables | Self-reported swallowing problem by T-SSQ (Yes): | Maximum tongue pressure (kPa): | Model 1 | Model 2 |
| | adjusted OR (95% CI) | adjusted β (95% CI) | | |
| Age (years old) | 1.05 (0.92, 1.18) | -0.65 (-1.00, -0.30)* | 1 (ref) | 0.98 (0.84, 1.14) |
| Sex (Female) | 0.55 (0.07, 4.38) | -1.00 (-5.76, 3.76) | 6.28 (0.44, 45.4) | 3.23 (0.45, 23.4) |
| Dependency status: Independent | 1 (ref) | 1 (ref) | 1 (ref) | 1 (ref) |
| Semi-dependent | 8.44 (0.89.30.1)† | -4.19 (-10.6, 2.28) | 0.21 (0.01, 6.26) | 0.80 (0.07, 8.65) |
| Dependent | 13.4 (1.24, 39.4)* | -6.67 (-14.1, 1.18)† | 22.3 (1.53, 53.3)* | 40.6 (3.30, 85.3)* |
| **Swallowing ability measures** | | | | |
| Self-reported swallowing problem (Yes) | | | 35.5 (3.49, 75.5)* | - |
| Maximum tongue pressure (<18 kPa) | | | - | 0.11 (0.01, 0.71)* |
| AUC (%) | | | 88.3 | 74.9 |

*p <0.05,

†p<0.10. OR, odds ratio; β, beta-coefficient; CI, confidence interval; ref, reference; AUC, Area under the Receiver Operating Characteristic (ROC) curve.

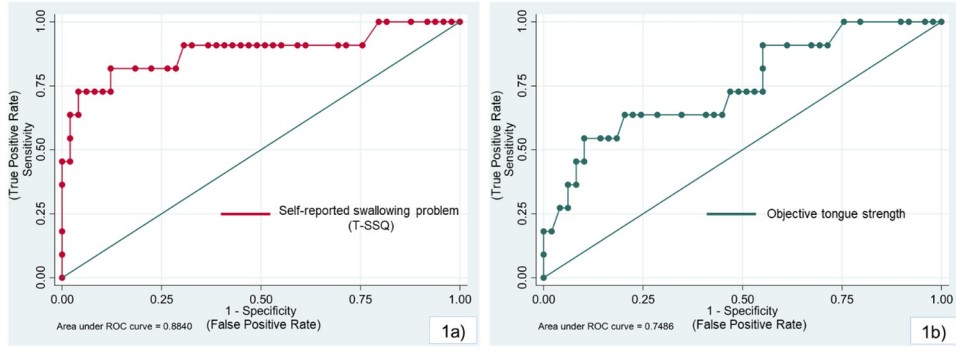

**Fig 1. Receiver Operating Curves (ROC) and % Area under the ROC curve (AUC) of the association between malnutrition risk and swallowing indices after adjusting for covariates.** 1a) Self-reported swallowing problem (T-SSQ), and 1b) Objective tongue strength (18-kPa cut-off value).

in estimating malnutrition risk was better than objective tongue strength. In this study, the standard EAT-10 was not used as a subjective swallowing index because some of our patients were unable to understand and complete the EAT-10 questionnaire. Due to its complexity and being time-consuming, our study introduced the T-SSQ for evaluating swallowing ability in older adults. The T-SSQ comprises only 4-item questions with a dichotomous answer, which is simpler than the 10-item questions answered using the 5-point Likert scale in the standard EAT-10.

Dependent status was significantly associated with low swallowing ability and malnutrition risk. Although the dependency level was associated with advanced age, a higher age was associated with lower tongue strength, but not having a swallowing problem. As supported by previous studies in healthy adults and older people, maximum tongue strength reduced with advanced age [20–22], which might be due to reduced musculoskeletal function [23] and masticatory muscle strength [20]. These findings imply that dependency status has a greater influence on swallowing ability than chronological age. Therefore, maintaining functional health and being active are important to prevent the progression of oral hypofunction in older adults.

The number of remaining teeth, posterior occlusal support, and denture type were not associated with the subjective or objective swallowing indices. Previous studies found that maximum tongue pressure increased with greater posterior occlusal support assessed using the Eicher index [20, 21]. In the present study, however, all edentulous patients wore a dental prosthesis when performing the tongue strength measurement because most of them required anterior denture teeth to position the pressure bulb. Wearing a dental prosthesis increases the number of posterior occlusal contacts, and therefore, enhances the bite force in edentulous individuals [24]. Individuals with higher occlusal forces present higher masticatory muscle strength [20], which is associated with lower dysphagia risk [15, 25]. Thus, wearing a dental prosthesis might reduce malnutrition risk in edentulous older adults regardless of the remaining functional teeth and posterior occlusal support.

**Table 3. PPV, NPV, sensitivity, and specificity (%) between swallowing indices and malnutrition risk.**

| Impaired swallowing ability | Malnutrition risk: (%) | | | |
|---|---|---|---|---|
| | PPV | NPV | Sensitivity | Specificity |
| Self-reported swallowing problem by TSSQ (Yes) | 71.4 | 87.2 | 45.5 | 96.0 |
| Low tongue strength: (< 18 kPa) | 40.0 | 86.0 | 36.4 | 87.8 |

In accordance with previous studies in middle-aged and older adults, malnutrition risk was associated with low tongue strength [3, 26]. To categorize low and high tongue strength, our study chose a cut-off value of 18 kPa because it gave the highest AUC value when plotting the ROC curve. The Japanese Society of Gerodontology suggests using 30 kPa as a cut-off value to diagnose decreased tongue strength [2], because a study in Japanese older adults reported that individuals with at least 30-kPa maximum tongue pressure could consume regular food [27]. Furthermore, a study in Canadian older adults in long-term care used a value of 26 kPa, the average tongue pressure of the study samples, as the cut-off value to categorize tongue pressure into low and high levels [3]. In our study, however, using either 30- or 26-kPa tongue pressure as a cut-off value gave relatively low sensitivity and PPV in estimating malnutrition risk, and resulted in a nonsignificant association with a subjective swallowing problem determined by the T-SSQ. However, due to the limited sample size in this pilot study, the 18-cut off value cannot yet be defined as representative of the Thai population. Because the thickness of the swallowing muscles might be different among ethnicities [28], individual studies may need to identify the ethnic-specific normal values of tongue strength.

The sensitivity of the subjective and objective swallowing indices in estimating malnutrition risk was comparable. However, the PPV value of the T-SSQ was about 1.8-fold greater than that of tongue strength. Moreover, the AUC obtained from the T-SSQ and malnutrition risk was 15.2% higher than the objective tongue strength value. These findings indicated that the T-SSQ might be a more appropriate tool for estimating malnutrition risk in older adults. As supported by earlier studies [11, 29], recognizing signs and symptoms with a thorough history taking is key in early diagnosis and detecting swallowing impairment. Based on our results, we suggest that tongue strength measurement could be a supplemental tool to confirm the subjective finding whenever patients or their caregivers have communication problems or are unaware of the symptoms.

Treating oral frailty and oral hypofunction requires a multidisciplinary approach. Thus, dentists can be part of a holistic team by early detection of declined swallowing function to prevent the progression into the irreversible dysfunction stage [2]. The present study suggests using the T-SSQ as a screening method for evaluating the swallowing ability in older adults that does not require an experienced physician in routine dental practice. In addition, we propose a concept for identifying a cut-off value to categorize lower and higher tongue strength using malnutrition risk as an outcome.

Some limitations were noted in this study. First, because all edentulous older adults in this study wore a dental prosthesis and visited the dental clinic for maintenance recall, the swallowing ability of the edentulous patients without a denture or with an ill-fitting denture was not evaluated. Second, the present pilot study did not include a control group for swallowing ability evaluation. The patients who were diagnosed with dysphagia by physicians as a positive control was not included because we wanted to develop a screening tool for swallowing impairment rather than a tool for dysphagia diagnosis. Third, due to the limited number of samples with 70% study power, the cut-off tongue strength value cannot yet be generalized to other populations. Therefore, further study in a larger population is required to verify the reliability, validity, and sensitivity of the questionnaire. Measurement equivalence of the T-SSQ should be performed in older adults with different cultural backgrounds. Convergent validity should be done with other standard subjective swallowing indexes, such as EAT-10 and objective tools, such as video fluoroscopy and fiberoptic endoscopic examination of swallowing [2, 30, 31]. Further use of the simplified questionnaire for early detection of swallowing problem in a clinic and community-based study by caregiver and non-healthcare personnel should be evaluated.

## Conclusions

Self-reported swallowing problems determined by the T-SSQ were associated with objective tongue strength, indicating convergent validity of the newly developed subjective index. Both subjective T-SSQ and objective tongue strength indices were associated with malnutrition risk in older adults. However, the subjective T-SSQ better estimated malnutrition risk than the objective tongue strength.

## Supporting information

**S1 File. Original nine common signs and symptoms of dysphagia.**
(PDF)

**S2 File. Questionnaire for data collection (Thai and English versions), Thai-version of Simplified Swallowing Questionnaire (T-SSQ), Thai-version of Mini-Mental State Evaluation (MMSE) (MMSE-Thai), and Thai-Mini Nutritional Assessment (MNA-Thai).**
(PDF)

**S3 File.**
(XLSX)

## Acknowledgments

The authors gratefully acknowledge Dr. Kevin Tompkins for language revision of the manuscript.

## Author Contributions

**Conceptualization:** Nareudee Limpuangthip, Orapin Komin.

**Data curation:** Nareudee Limpuangthip, Orapin Komin, Teerawut Tatiyapongpaiboon.

**Formal analysis:** Nareudee Limpuangthip.

**Funding acquisition:** Nareudee Limpuangthip.

**Investigation:** Teerawut Tatiyapongpaiboon.

**Methodology:** Nareudee Limpuangthip, Orapin Komin, Teerawut Tatiyapongpaiboon.

**Project administration:** Orapin Komin.

**Resources:** Orapin Komin.

**Software:** Nareudee Limpuangthip.

**Supervision:** Orapin Komin.

**Validation:** Nareudee Limpuangthip, Orapin Komin.

**Visualization:** Orapin Komin.

**Writing – original draft:** Nareudee Limpuangthip, Orapin Komin.

**Writing – review & editing:** Nareudee Limpuangthip, Orapin Komin, Teerawut Tatiyapongpaiboon.

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
