## [Decision Letter · Decision Letter 0]

12 Aug 2021

PONE-D-21-21718

A simplified method for evaluating swallowing ability and estimating malnutrition risk in older adults

PLOS ONE

Dear Dr. Komin,

Thank you for submitting your manuscript to PLOS ONE. After careful consideration, we feel that it has merit but does not fully meet PLOS ONE’s publication criteria as it currently stands. Therefore, we invite you to submit a revised version of the manuscript that addresses the points raised during the review process.

We look forward to receiving your revised manuscript.

Kind regards,

Pravinkumar G. Patil

Academic Editor

PLOS ONE

Journal Requirements:

Reviewers' comments:

Reviewer's Responses to Questions

**Comments to the Author**

1. Is the manuscript technically sound, and do the data support the conclusions?

Reviewer #1: Partly

Reviewer #2: Partly

2. Has the statistical analysis been performed appropriately and rigorously? 

Reviewer #1: No

Reviewer #2: Yes

3. Have the authors made all data underlying the findings in their manuscript fully available?

Reviewer #1: Yes

Reviewer #2: No

4. Is the manuscript presented in an intelligible fashion and written in standard English?

Reviewer #1: Yes

Reviewer #2: No

5. Review Comments to the Author

Reviewer #1: The authors have embarked on a novel method to identify a screening tool for evaluating swallowing ability in older adults. The manuscript is well written and systematically presented.

However, the conclusions made from this study needs further justification for application on a larger population because its reliability and validity is yet unclear. Probably, this can be introduced as a pilot study to further analyze on the sensitivity of the proposed 4 items self reported assessment tool. Things tat need further clarification include the selection of the items, independence of each item from the other, the scoring value adopted for this tool, the measurement equivalence of the tool, etc. The authors do not mention in which language the questionnaire was administered; if in the local language, any translation was done and cross-culturally validated. The cut off value for the Objective Swallowing Assessment was way below that reported in cited literature and in case that cut-off is raised there is a statistically significant variation between the Objective and Subjective Swallowing tools applied in this study. The sample size in this study needs to be evaluated if is adequate to introduce a sensitive assessment tool that can be used as a key instrument in early diagnosis and detecting of swallowing impairment among older adults.

Reviewer #2: Control group should have been included. Newly developed questionnaire should be validated. Statistics data should be furnished completely. Relevant figures with legends to be provided. copy of questionnaire should be given.

6. PLOS authors have the option to publish the peer review history of their article (what does this mean?). If published, this will include your full peer review and any attached files.

Reviewer #1: No

Reviewer #2: No

---

## [Author Response · Author response to Decision Letter 0]

30 Aug 2021

Response to reviewers

We are pleased to submit our revised manuscript Number: PONE-D-21-21718. The title has been modified to ‘A simplified method for evaluating swallowing ability and estimating malnutrition risk: A pilot study in older adults.’ The newly developed questionnaire has been named ‘Thai-version of Simplified Swallowing Questionnaire’ or ‘T-SSQ’. The requested revisions have been made in the manuscript in track changes, and our point-by-point responses are below.

Reviewer No. 1: 

Reviewer point #1: The authors have embarked on a novel method to identify a screening tool for evaluating swallowing ability in older adults. The manuscript is well written and systematically presented.

However, the conclusions made from this study needs further justification for application on a larger population because its reliability and validity is yet unclear. Probably, this can be introduced as a pilot study to further analyze on the sensitivity of the proposed 4 items self-reported assessment tool. 

Author response #1: Thank you for your comment. We have revised our study limitations, further study suggestions and conclusion. Due to a limitation, the study has been changed into a pilot study. 

Reviewer point #2: Things that need further clarification include the selection of the items, independence of each item from the other, the scoring value adopted for this tool, the measurement equivalence of the tool, etc. 

Author response #2: Development of the T-SSQ including the selection of the items, the scoring value adopted for the T-SSQ, its interpretation, and its validity testing have been clarified in the Materials and Methods section (Phase I subsection, Page 5-6). We did not use any statistical test to evaluate the independence of each item from the other because the items were selected by experts.

Reviewer point #3: The authors do not mention in which language the questionnaire was administered; if in the local language, any translation was done and cross-culturally validated. 

Author response #3: Thai language was used for the swallowing questionnaire. The cross-cultural translation from English to Thai version was performed, and the descriptions have been added in the Materials and Methods section (Phase I subsection, Page 5)

Reviewer point #4: The cut off value for the Objective Swallowing Assessment was way below that reported in cited literature and in case that cut-off is raised there is a statistically significant variation between the Objective and Subjective Swallowing tools applied in this study. 

Author response #4: The descriptions about the cut-off value have been revised in the Discussion section (Page 15-16) according to the reviewer’s suggestion. 

Reviewer point #5: The sample size in this study needs to be evaluated if is adequate to introduce a sensitive assessment tool that can be used as a key instrument in early diagnosis and detecting of swallowing impairment among older adults.

Author response #5: The power analysis of the sample size has been revised by evaluating whether the T-SSQ was a sensitive assessment tool for using as a key instrument in early diagnosis and detecting swallowing impairment among older adults. The revisions have been made in the Materials and Methods section (Power analysis subsection, Page 9)

Reviewer No. 2:

Reviewer point #1: Control group should have been included. 

Author response #1: Our pilot study did not include patients who were diagnosed with dysphagia by physicians. Therefore, a positive control group was not present in this study. This was because we wanted to develop a screening tool for swallowing ability impairment rather than a tool for dysphagia diagnosis. This limitation has been added in the Discussion section (Page 17).

Reviewer point #2: Newly developed questionnaire should be validated. Statistics data should be furnished completely. 

Author response #2: Descriptions about the validation of the newly developed questionnaire (T-SSQ) has been added in the Materials and Methods section (Phase I subsection). The statistical analysis has also been revised. 

Reviewer point #3: Relevant figures with legends to be provided. Copy of questionnaire should be given. 

Author response #3: The relevant figures with legends have been provided in the manuscript file. A copy of the questionnaire (in Thai, and English translation) has been provided as a supplementary file.

Additional responses to the comments in the attached Pdf files:

Materials and Methods section

- The references of the Thai-version of the Mini-Nutritional Assessment (MNA) has been revised.

- The references for the tongue pressure measurement and the Thai-version of Mini-Mental State Evaluation (MMSE) have been added.

- To evaluate the reliability of the T-SSQ in older adults, the inter-examiner reliability was examined in 15 patients at the patients’ first evaluation visit. Test-retest reliability was evaluated by reinterviewing these patients one week later. The descriptions have been revised in the Swallowing ability assessment subsection (Page 8).

Results section

- The results for the descriptive statistics have been demonstrated in the Result section and Tables.

- The statistical tests have been added as a footnote of Table 1.

Discussion section

- The patients who were unable to perform a tongue pressure test due to severely declined functional or intellectual conditions were excluded. This exclusion criterion has been included in the Materials and Methods section (Page 5).

- The study limitations have been added according to the reviewers’ suggestions.

Sincerely yours

Orapin Komin

Corresponding author

---

## [Decision Letter · Decision Letter 1]

6 Oct 2021

PONE-D-21-21718R1A simplified method for evaluating swallowing ability and estimating malnutrition risk: A pilot study in older adultsPLOS ONE

Dear Dr. Orapin Komin,

Thank you for submitting your manuscript to PLOS ONE. After careful consideration, we feel that it has merit but does not fully meet PLOS ONE’s publication criteria as it currently stands. Therefore, we invite you to submit a revised version of the manuscript that addresses the points raised during the review process.

ACADEMIC EDITOR: There are minor revisions needed and have been mentioned in the attached file. Please address those carefully and resubmit the revised manuscript.

We look forward to receiving your revised manuscript.

Kind regards,

Pravinkumar G. Patil

Academic Editor

PLOS ONE

Journal Requirements:

Additional Editor Comments (if provided):

There are minor revisions needed and have been mentioned in the attached file. Please address those carefully and resubmit the revised manuscript.

Reviewers' comments:

Reviewer's Responses to Questions

**Comments to the Author**

1. If the authors have adequately addressed your comments raised in a previous round of review and you feel that this manuscript is now acceptable for publication, you may indicate that here to bypass the “Comments to the Author” section, enter your conflict of interest statement in the “Confidential to Editor” section, and submit your "Accept" recommendation.

Reviewer #1: All comments have been addressed

Reviewer #2: All comments have been addressed

2. Is the manuscript technically sound, and do the data support the conclusions?

Reviewer #1: Yes

Reviewer #2: Yes

3. Has the statistical analysis been performed appropriately and rigorously? 

Reviewer #1: Yes

Reviewer #2: Yes

4. Have the authors made all data underlying the findings in their manuscript fully available?

Reviewer #1: Yes

Reviewer #2: Yes

5. Is the manuscript presented in an intelligible fashion and written in standard English?

Reviewer #1: Yes

Reviewer #2: Yes

6. Review Comments to the Author

Reviewer #1: The corresponding author has addressed all comments to satisfaction and the revised version of the manuscript fulfils all basic requirements set forward by the journal editorial team.

Reviewer #2: all the comments in the previous review have been addressed properly and diligently. Limitations of the study have added in the manuscript. Content has been concised and made clearer to understand for readers.

7. PLOS authors have the option to publish the peer review history of their article (what does this mean?). If published, this will include your full peer review and any attached files.

Reviewer #1: No

Reviewer #2: No

---

## [Author Response · Author response to Decision Letter 1]

9 Oct 2021

Response to reviewers

We are pleased to submit our revised manuscript Number: PONE-D-21-21718R1, entitle ‘A simplified method for evaluating swallowing ability and estimating malnutrition risk: A pilot study in older adults.’ The requested revisions have been made in the manuscript in track changes, and our point-by-point responses are below.

Reviewers: 

Reviewer point #1: Introduction section

Author response #1: 

- The word ‘decreased occluding teeth’ has been revised to ‘decreased occluding pairs of natural teeth’ 

- The word ‘reduced chewing function’ has been revised to ‘declined masticatory function’

Reviewer point #2: Materials and Methods section. 

Author response #2: 

- Study design and participants subsection (Page 5): The sentence has been revised to ‘The participants were older adults (aged ≥ 60 years old) recruited from patients who received dental treatment at the Geriatric and Special Patients Care Clinic, Faculty of Dentistry, Chulalongkorn University during August 2017– January 2019 (total duration of 1 year and 5 months).’ 

- Covariates subsection (Page 8): The sentence has been revised to ‘With a score ranging from 0–30, the participants were considered as having mild cognitive impairment (MCI) when the score was below 18 and below 22 when they had attained at least primary and above primary education, respectively.’

Reviewer point #3: Discussion section

- Page 14: The sentence has been revised to ‘The T-SSQ comprises only 4-item questions with a dichotomous answer, which is simpler than the 10-item questions answered using the 5-point Likert scale in the standard EAT-10.’

- Page 16: The limitations of the study have been rephrased. 

Sincerely yours

Orapin Komin

Corresponding author

---

## [Decision Letter · Decision Letter 2]

31 Jan 2022

A simplified method for evaluating swallowing ability and estimating malnutrition risk: A pilot study in older adults

PONE-D-21-21718R2

Dear Dr. Orapin Komin,

We’re pleased to inform you that your manuscript has been judged scientifically suitable for publication and will be formally accepted for publication once it meets all outstanding technical requirements.

Kind regards,

Pravinkumar G. Patil

Academic Editor

PLOS ONE

Additional Editor Comments (optional):

Reviewers' comments:

Reviewer's Responses to Questions

**Comments to the Author**

1. If the authors have adequately addressed your comments raised in a previous round of review and you feel that this manuscript is now acceptable for publication, you may indicate that here to bypass the “Comments to the Author” section, enter your conflict of interest statement in the “Confidential to Editor” section, and submit your "Accept" recommendation.

Reviewer #2: All comments have been addressed

2. Is the manuscript technically sound, and do the data support the conclusions?

Reviewer #2: Yes

3. Has the statistical analysis been performed appropriately and rigorously? 

Reviewer #2: Yes

4. Have the authors made all data underlying the findings in their manuscript fully available?

Reviewer #2: Yes

5. Is the manuscript presented in an intelligible fashion and written in standard English?

Reviewer #2: Yes

6. Review Comments to the Author

Reviewer #2: All the corrections suggested during revision 1 & revision 2 are done satisfactorily. The data is represented in simplified manner as compared to the original version. Queries regarding statistics are clarified in revision 2. Content is fine from grammatical point of view.

7. PLOS authors have the option to publish the peer review history of their article (what does this mean?). If published, this will include your full peer review and any attached files.

Reviewer #2: No

---

## [Editor Report · Acceptance letter]

7 Feb 2022

PONE-D-21-21718R2 

A simplified method for evaluating swallowing ability and estimating malnutrition risk: A pilot study in older adults 

Dear Dr. Komin:

I'm pleased to inform you that your manuscript has been deemed suitable for publication in PLOS ONE. Congratulations! Your manuscript is now with our production department. 

Kind regards, 

on behalf of

Dr. Pravinkumar G. Patil 

Academic Editor

PLOS ONE